# SARS-CoV-2 RBD Conjugated to Polyglucin, Spermidine, and dsRNA Elicits a Strong Immune Response in Mice

**DOI:** 10.3390/vaccines11040808

**Published:** 2023-04-06

**Authors:** Ekaterina A. Volosnikova, Iuliia A. Merkuleva, Tatiana I. Esina, Dmitry N. Shcherbakov, Mariya B. Borgoyakova, Anastasiya A. Isaeva, Valentina S. Nesmeyanova, Natalia V. Volkova, Svetlana V. Belenkaya, Anna V. Zaykovskaya, Oleg V. Pyankov, Ekaterina V. Starostina, Alexey M. Zadorozhny, Boris N. Zaitsev, Larisa I. Karpenko, Alexander A. Ilyichev, Elena D. Danilenko

**Affiliations:** State Research Center of Virology and Biotechnology VECTOR, Rospotrebnadzor, 630559 Koltsovo, Russia

**Keywords:** SARS-CoV-2, COVID-19, receptor-binding domain, RBD, adjuvant

## Abstract

Despite the rapid development and approval of several COVID vaccines based on the full-length spike protein, there is a need for safe, potent, and high-volume vaccines. Considering the predominance of the production of neutralizing antibodies targeting the receptor-binding domain (RBD) of S-protein after natural infection or vaccination, it makes sense to choose RBD as a vaccine immunogen. However, due to its small size, RBD exhibits relatively poor immunogenicity. Searching for novel adjuvants for RBD-based vaccine formulations is considered a good strategy for enhancing its immunogenicity. Herein, we assess the immunogenicity of severe acute respiratory syndrome coronavirus 2 RBD conjugated to a polyglucin:spermidine complex (PGS) and dsRNA (RBD-PGS + dsRNA) in a mouse model. BALB/c mice were immunized intramuscularly twice, with a 2-week interval, with 50 µg of RBD, RBD with Al(OH)_3_, or conjugated RBD. A comparative analysis of serum RBD-specific IgG and neutralizing antibody titers showed that PGS, PGS + dsRNA, and Al(OH)_3_ enhanced the specific humoral response in animals. There was no significant difference between the groups immunized with RBD-PGS + dsRNA and RBD with Al(OH)_3_. Additionally, the study of the T-cell response in animals showed that, unlike adjuvants, the RBD-PGS + dsRNA conjugate stimulates the production of specific CD4+ and CD8+ T cells in animals.

## 1. Introduction

Most vaccines for COVID-19 prevention aim to elicit an immune response to the severe acute respiratory syndrome coronavirus 2 (SARS-CoV-2) surface S-protein or its receptor-binding domain (RBD). The choice of RBD seems obvious since RBD plays a key role in target cell recognition and virus entry into the cell [1]. Since the announcement of a global COVID-19 pandemic, there has been a continuous evolution of SARS-CoV-2, accelerating after about a year, leading primarily to a change in its antigenic properties [2]. This circumstance reduces or completely cancels the effectiveness of vaccines developed at the first stage, using the sequences of the original virus (Wuhan strain) [3]. However, along with the work on updating the antigenic composition, work is needed to improve vaccine platforms to increase their safety and increase efficiency. Current vaccines include both traditional as well as modern vaccine platforms, including viral vectors or DNA, and mRNA-based vaccines. Despite the demonstrated efficacy, vaccination using viral vectors—such as DNA and RNA vaccines—has been associated with numerous unintended effects, such as vaccine-associated myocarditis/pericarditis, thrombotic events, anaphylaxis and allergy-like reactions, and delayed hypersensitivity reactions [4,5,6,7,8]. Experts point out that in the case of DNA and RNA vaccines, these risks are minimal, and the benefits of vaccination outweigh them. The final answer to this question can, perhaps, be given only after the completion of long-term observations. In this regard, subunit vaccines based on recombinant RBD or S-protein are a more proven approach.

We previously assessed the immunogenicity of recombinant RBD supplemented with aluminum hydroxide adjuvant in five animal models (mice, rabbits, ferrets, hamsters, and chickens) and showed that adjuvanted RBD elicited a strong humoral immune response in animals [9]. However, although aluminum salt adjuvants are considered relatively safe, they may cause some side effects in humans, especially in patients with kidney disease [10]. In addition to adjuvants, various carriers, antigen multimerization strategies, etc. are used to enhance immunogenicity [11]. The immunogenicity of multimeric forms of the antigen is increased due to B-cell receptor crosslinking, increased avidity of multimeric proteins, and enhanced retention of nanoparticles larger than 20 nm in the lymph nodes. Multivalent variants of RBD have been shown to be more effective in inducing an immune response than monomeric variants in laboratory animals [12,13,14,15,16].

In this study, we combined the idea of safe natural adjuvants and multimerization, and developed RBD conjugates with a polyglucin:spermidine complex (PGS) and double-stranded RNA (dsRNA) isolated from a killer strain of *Saccharomyces cerevisiae* 448 Y yeast. Polyglucin (high-molecular-weight dextran) is one of the best-studied α-glucans, which is used for the delivery of drugs and immunogens. Polyglucin provides prolonged release of the active substance and is also an immunomodulator inducing humoral and cellular immunity [17,18]. Dextran can directly interact with the DC-SIGN cellular receptor, mediating phagocytosis. It is an inexpensive reagent, has a long history of medical usage, and is generally recognized as safe (GRAS) by the FDA [18,19]. In addition, its structure, particularly its hydroxyl groups, makes it easy to functionalize and chemically modify dextran [20].

Earlier, we modified dextran with spermidine and used the PGS complex for nucleic acid delivery. Positively charged spermidine ensures the interaction between polyglucin:spermidine conjugates and negatively charged nucleic acid molecules. The ability of natural polyamines, including spermidine, to self-assemble has been studied in cells by the example of aggregates of polyamines and genomic DNA, which protect it against degradation by nucleases [21]. Spermidine is found to be associated with nucleic acids in almost all tissues; it is a cation at all pH values and stabilizes cell membranes and nucleic acids. Spermidine per se exerts an immunomodulatory effect and is of interest in developing drugs for treating infectious diseases, cancer, and immunodeficiency conditions (DrugBank: DB03566 (Edmonton, Alberta, Canada), https://go.drugbank.com/drugs/DB03566 (accessed on 30 January 2023)). In the present work, the nucleic acid was represented by the natural adjuvant dsRNA rather than a plasmid. Double-stranded RNAs (dsRNAs) have long been known as regulators of innate immunity and interferonogenesis inducers. The double-stranded RNA are foreign to mammalian cells and naturally represent either a viral genome or products of the viral replicative cycle. dsRNA is recognized by the Toll-like receptor 3 (TLR3) cellular receptor. The receptor is expressed mainly by cells of the immune system: dendritic cells, macrophages, natural killers, and T-lymphocytes; it is also found on somatic cells. In most cell types, TLR3 is located inside the cell compartments: on the membranes of the endoplasmic reticulum, endosomes, and lysosomes, although sometimes it is found on the cell membrane. As a result of binding to dsRNA, the receptor is dimerized and triggers a cascade of intracellular signaling pathways and activation of type 1 interferon (IFN) genes, as well as a number of pro-inflammatory cytokines. In turn, the IFN produced by cells repeatedly enhances the transcription of genes encoding IFN-dependent enzymes that directly implement the antiviral response [22,23].

Taking into consideration the properties of all the components, we assumed that the conjugate of RBD-PGS combined with dsRNA might have a number of advantages: the multimerized RBD exposed on the particle surface will enhance the B-cell response, and dsRNA released after the particle penetration into cells can enable the induction of the innate and cellular immune responses.

The aim of this study was to evaluate the immunogenicity of the conjugate of RBD-PGS combined with dsRNA in the BALB/c mice model.

## 2. Materials and Methods

### 2.1. Plasmid Construction, Recombinant Protein Expression, and Purification

The RBD and S-trimer were prepared as described previously [24]. The Wuhan-1 strain spike nucleotide sequence (GenBank: MN908947) was codon-optimized and synthesized. The RBD coding fragment (308 V–542 N a.a.) was amplified and cloned into the pVEAL2 transposon plasmid in frame with the N-terminal spike signal sequence (MFVFLVLLPLVSSQC) and the C-terminal 10 × His-tag.

CHO-K1 cells were transfected with the pVEAL2-S-RBD and helper plasmid pCMV (CAT) T7-SB100, encoding SB100 transposase, using Lipofectamine 3000 (Invitrogen, Carlsbad, CA, USA). The transfected cells were selected with puromycin (10 µg/mL), and high-producing clones were obtained by dilution cloning and cultured in roller bottles at 37 °C on DMEM/F-12 (1:1) medium supplemented with 2% FBS and 50 µg/mL gentamicin.

The RBD protein from CHO-K1 culture medium was purified by Ni-NTA and ion-exchange chromatography and analyzed by SDS-PAGE in a 15% separating polyacrylamide gel. The RBD samples were dialyzed against PBS and sterilized by filtration through 0.22 µm filters.

The S-protein 1M-P1213 coding gene fragment was designed with a removed protease cleavage site, K986P and V987P amino acid stabilizing substitutions, and a C-terminal T4 bacteriophage fibritin trimerization domain followed by a His-tag. The resulting DNA fragment was cloned into the pVEAL2 vector and produced using the CHO-K1 cell line as described above.

### 2.2. Synthesis of Polyglucin: Spermidine-RBD (PGS-RBD) Conjugates

In order to obtain the PGS-RBD conjugates, first, dextran was activated by oxidation with sodium periodate: 1 mole of dextran 40,000 (MP Biomedicals, Irvine, CA, USA) per 40 moles of sodium periodate. The mixture was then incubated for 60 min, and the remaining oxidizing agent was removed from the activated dextran by gel filtration on a column packed with Sephadex G-75 equilibrated with 50 mM carbonate buffer (pH 8.6). RBD protein was then added to the activated dextran at the ratio of 1 mole of dextran per 1 mole of protein, and the mixture was incubated for 2 h. Spermidine was added to the mixture at a ratio of 1 mole of dextran per 30 moles of spermidine, and the mixture was incubated for 2 h. Sodium borohydride was then added to the mixture at a ratio of 80 moles of borohydride per 1 mole of dextran, with further incubation for 2 h. The unreacted components were separated from the mixture by gel filtration on a column packed with Sephadex G-25 equilibrated with PBS. The resulting conjugates were sterile-filtered through 0.22 µm filters.

### 2.3. RBD-PGS + dsRNA Particle Assembly

To assemble the particles, the resulting RBD-PGS conjugates were mixed with the dsRNA extracted from *Saccharomyces cerevisiae* Y 448 killer strain (1:10 by weight) and incubated for 1 h. at 2–8 °C. The effectiveness of the formation of RBD-PGS + dsRNA complexes was evaluated by RNA and protein electrophoretic mobility shift assays.

To evaluate the size and shape of the resulting particles, their suspensions were applied to copper grids for electron microscopy, covered with a carbon-stabilized formvar film. The preparations were stained with a 2% aqueous solution of phosphotungstic acid and examined using a JEM-1400 electron microscope (JEOL, Akishima, Tokyo, Japan). A Veleta digital camera (SIS, Schwentinental, Germany) was used to acquire images, and the iTEM software package (SIS, Schwentinental, Germany) was used to analyze and process the images.

Gel filtration of the assembled complexes was carried out on a 10 mL Sepharose CL-6B column in PBS (pH 7.4). The samples of the studied preparations were applied in equimolar amounts to the nucleotide material.

### 2.4. Dynamic and Electrophoretic Light Scattering

The hydrodynamic size of nanoparticles formed by dsRNA and PGS-RBD and their kinetic layer potential (zeta potential) were determined by dynamic and electrophoretic light scattering methods using a Zetasizer NanoZSPlus (Malvern Instruments; Malvern, UK). ZEN0040 cuvettes were used to measure the size, and DTS1070 was used to measure the possibility of the kinetic layer of nanoparticles. All measurements were made in triplicate at 25 °C. The mean particle size parameter, polydispersity index (PdI) parameter, and mean zeta ability were used.

### 2.5. Animal Immunization

All the experimental protocols and procedures were approved by the Bioethics Committee of the State Research Center of Virology and Biotechnology Vector (SRC VB Vector/September 10, 2020, approved by the protocol of Bioethics Committee No. 5 as of 1 October 2020).

Groups of BALB/c mice (*n* = 10 animals per group) were maintained in separate cages under standard conditions and had free access to water and food at all times.

BALB/c mice were immunized intramuscularly twice, with a 2-week interval, with 50 µg of RBD, RBD-PGS conjugates, RBD-PGS + dsRNA particles, or RBD mixed with Al(OH)_3_. Blood and spleen samples were obtained two weeks after immunization. Blood was incubated for 1 h. at 37 °C and for 2 h at 4 °C, then centrifuged at 7000× *g* for 10 min to separate cellular elements. The sera were deactivated by heating at 56 °C for 30 min and stored at −20 °C.

Spleens were sequentially minced on nylon filters for cells with pore diameters of 70 and 40 µm (BD Falcon, Franklin Lakes, NJ, USA). After lysis of erythrocytes with lysis buffer (Sigma, Burlington, MA, USA), splenocytes were washed twice in complete RPMI medium and placed in 1 mL of RPMI medium with 2 mM L-glutamine, 50 µg/mL gentamicin, and 10% FBS (Thermo Fisher Scientific, NY, USA). Cells were counted using a TC20 automatic cell counter (Bio-Rad, Hercules, CA, USA). Among the splenocytes, the number of IFN-γ-producing cells was determined using the ELISpot and ICS methods.

### 2.6. ELISAs

The 96-well plates (Corning, Glendale, AZ, USA) were coated with 100 µL of recombinant RBD or S-trimer (1 µg/mL) in 2 M urea in PBS overnight at 4 °C. The plates were then washed with PBST (PBS supplemented with 0.05% Tween-20) and blocked with a blocking buffer (PBST supplemented with 1% of casein) for 1.5 h. at RT. Mouse sera were serially diluted and added to the blocked plates and incubated at RT for 1 h. The plates were then washed with PBST and incubated with goat anti-mouse IgG-HRP secondary antibodies (Sigma–Aldrich, St. Louis, MO, USA) at RT for 1 h. The plates were washed three times with PBST, and the HRP substrate TMB (Amresco, Solon, OH, USA) was added. The reactions were stopped using 1 N HCl. Absorbance was measured at 450 nm using a Varioskan Lux multimode microplate reader (Thermo Fisher Scientific, Waltham, MA, USA).

The endpoint titer for each serum was defined as the maximum dilution at which a positive result was obtained (>median + 3 × SD of the ODs of the negative controls).

### 2.7. Viral Neutralization Test

The serum titers of SARS-CoV-2 neutralizing antibodies were determined through cytopathic effect (CPE) inhibition assays.

Vero E6 cells were seeded in 96-well tissue culture plates (Thermo Fisher Scientific, Waltham, MA, USA) and cultured for 24 h. to form monolayers. Serial twofold dilutions of serum samples (1/10 to 1/5120) were mixed with equal volumes of virus suspension containing the 100 TCID50 SARS-CoV-2 coronavirus strain nCoV/Victoria/1/2020 (obtained from the State Collection of Causative Agents of Viral Infections and Rickettsioses, SRC VB Vector, Novosibirsk region, Russia). The mixture was incubated at 37 °C for 1 h before being added to Vero E6 cells. Four days post infection, the cells were stained with 0.2% gentian violet solution. The presence of a specific cytopathic effect was assessed visually through microscopic examination.

The neutralizing antibody titers were defined as the dilutions of serum that completely prevented the CPE in 50% of the wells.

### 2.8. IFN-γ ELISpot

Analysis of the T-cell immune response was performed by using the Mouse IFN-gamma ELISpot Kit (R&D Systems, Minneapolis, MN, USA) according to the manufacturer’s instructions. Splenocytes were seeded (5 × 10^5^ cells/well, RPMI medium with 10% FBS) and stimulated with a pool of peptides (20 μg/mL each) from the RBD of SARS-CoV-2 S protein sequence, restricted by major histocompatibility complex (MHC) class I (H2-Dd, H-2-Kd, and H-2-Ld) and MHC class II (H2-IAd and H2-IEd) molecules of BALB/c mice. The peptides were selected using the IEDB Analysis Resource instruments and synthesized by AtaGenix Laboratories (Wuhan, China). For the negative control, splenocytes were incubated without stimulation, and Concanavalin A was added to the well for the positive control. The cells were incubated for 20 h at 37 °C in the presence of 5% CO_2_. Subsequently, the plates were washed and incubated with the primary antibody against IFN-γ. The plates were washed again and incubated with a secondary antibody conjugated to alkaline phosphatase. Finally, the plates were washed again and incubated with BCIP/NBT. The number of IFN-γ-producing cells was counted using an ELISpot reader (Carl Zeiss, Oberkochen, Germany). The number of spot-forming units (SFU) per million cells was calculated by subtracting the average value from the negative control wells.

### 2.9. Intracellular Cytokine Staining

Splenocytes were plated at 2 × 10^6^ cells/well into the 24-well culture plates (TPP, Trasadingen, Switzerland) and stimulated with the pool of peptides mentioned above. Cells were incubated for 4 h at 37 °C in the presence of 5% CO_2_ and for an additional 16 h with Brefeldin A (5 μg/mL, BD Biosciences, CA, USA). The next day, the cells were stained with anti-CD3 conjugated to Alexa Fluor 700 (BD), anti-CD4 conjugated to BV786 (BD), and anti-CD8 conjugated to FITC (BD); fixed with 1% paraformaldehyde in PBS; and permeabilized with 0.5% Tween-20 in PBS according to the manufacturer’s instructions. Then, the cells were stained with anti-IFN-γ APC (BD Biosciences, CA, USA) and analyzed using a ZE5 flow cytometer (Bio-Rad, Hercules, CA, USA) and the Everest software.

### 2.10. Statistical Analysis

All statistical analyses were performed using GraphPad Prism 8.0 software, with *p* < 0.05 considered to indicate statistical significance. The statistical significance among different animal groups was determined using a two-tailed nonparametric Mann–Whitney U test with a 95% confidence interval or the Kruskal–Wallis test (for more than two groups).

## 3. Results

### 3.1. Preparation and Characterization of RBD Conjugated to Polyglucin:Spermidine and dsRNA

We obtained RBD-PGS + dsRNA particles, the complexes consisting of dsRNA extracted from *S. cerevisiae* Y488 wrapped within the PGS envelope, with RBD molecules exposed on their surface (Figure 1A). For this purpose, RBD molecules were covalently bound to dextran in complex with spermidine. As one can see in the electropherogram (Figure 1B), the RBD-PGS conjugate, unlike the RBD protein, is visualized as a “cloud”. Then, dsRNA of *S. cerevisiae* that packed into the RBD-PGS envelope due to its interaction with positively charged spermidine was added to the obtained conjugates. The formation of RBD-PGS + dsRNA particles was implicitly confirmed by the slowing down of their electrophoretic mobility compared to that of the intact dsRNA (Figure 1C).

Analysis using the dynamic light scattering method showed that the polydispersity index (PdI) of the RBD-PGS + dsRNA particles was 0.19 ± 0.08. The average hydrodynamic diameter of RBD-PGS + dsRNA was 55 ± 5 nm. Evaluation of the zeta potential of the complexes showed that the value of the zeta potential of the particles was −1.02 ± 0.36 mV, which is in agreement with our calculations. The presence of a weak charge may indicate that the positively charged cationic polymer almost completely packed the negatively charged dsRNA molecules.

We also recorded electron microscopy images of the complexes. As expected, the particles had a spherical shape and were ~30 nm in diameter.

### 3.2. Humoral Immune Response

Next, we immunized the groups of BALB/c mice with samples of RBD-PGS, RBD-PGS + dsRNA, or RBD with Al(OH)_3_ to evaluate the contribution of the adjuvants to the immunogenicity of the samples. The control groups received injections of PBS or RBD protein without any adjuvants.

Comparative analysis of titers of RBD- and S-specific IgG and neutralizing antibodies in mouse sera showed that PGS, PGS + dsRNA, and Al(OH)_3_ enhance a specific humoral immune response in animals. The geometric mean titer (GMT) of RBD-specific IgG in the group immunized with RBD was 20-, 45- and 60-fold less than in the RBD-PGS, RBD-PGS + dsRNA, and RBD + Al(OH)_3_ groups, respectively (Figure 2A); and the geometric mean titer of S-specific IgG in the group immunized with RBD was, on average, two orders less than in groups immunized with RBD with adjuvants (Figure 2B). There was no significant difference between the groups immunized with RBD-PGS + dsRNA and RBD with Al(OH)_3_ in ELISAs. However, the geometric mean titer of RBD-specific IgG was 2.5-fold (4.2 × 10^5^ vs. 1.7 × 10^5^) higher in the group immunized with RBD-PGS + dsRNA than that in the group immunized with RBD+PGS (Figure 2A,C).

The results of virus neutralization analysis correlate with the results of ELISA: the sera of animals immunized with the RBD-PGS + dsRNA and RBD + Al(OH)_3_ showed the highest activity (with reciprocal GMT of 600) in inhibiting the cytopathic effect of the virus. That’s about 6- and 60-fold higher than in the groups immunized with RBD-PGS and RBD, respectively.

### 3.3. Cellular Immune Response

The ELISpot analysis detecting the IFN-γ-specific secretion in mice splenocytes showed no statistically significant differences between the control group immunized with PBS and the groups that received RBD, RBD with Al(OH)_3_, or RBD-PGS. At the same time, mice immunized with RBD-PGS + dsRNA evoked a specific cellular immune response in animals (Figure 3A). Further, in this group, specific immune responses in populations of CD4+ and CD8+ T cells were determined using intracellular cytokine staining (ICS) followed by flow cytometry (Figure 3B). The analysis showed the presence of both CD4+ and CD8+ T cells specific for RBD epitopes in mice immunized with RBD-PGS + dsRNA with a predominance of CD4+ T cells (Figure 3B and Appendix A).

## 4. Discussion

Although a number of vaccines based on the full-length S-protein of SARS-CoV-2 had been approved in the beginning of the COVID-19 pandemic, RBD-based vaccines may be the first choice [25]. By now, at least seven RBD-based vaccines have been approved for use.

The application of RBD as a vaccine antigen makes it possible to achieve the induction of high titers of neutralizing antibodies since the main epitopes of neutralizing antibodies are located exactly in the RBD region. It has been shown that the use of the S-trimer or S1-fragment yields similar or worse results [26,27,28]. The absence of the effect of antibody-dependent enhancement (ADE) of infection observed in animals immunized with the full-length S-protein indicates a safer profile of RBD-based vaccines [29,30]. Finally, the RBD is more easily expressed and, unlike the full-length S-protein, can be obtained in larger quantities using various expression systems, such as insect cells, mammalian cells, yeast, plants, and *E. coli* [24,31,32].

However, due to its small size, RBD per se exhibits relatively weak immunogenicity. Therefore, various adjuvants and multivalent antigen display systems have been proposed to increase the effectiveness of RBD-based vaccines.

Using monomeric RBD as a vaccine antigen is indeed the simplest approach. In general, it was shown that the monomeric versions of the antigen supplemented with various adjuvants induced fairly good production of neutralizing antibodies in animal and human models; however, the T-cell response was undetectable or very weak and manifested itself by the presence of only CD4+ T cells [33,34,35].

It was later shown that the multivalent versions of RBD elicit an immune response in laboratory animals more efficiently compared to the monomeric versions.

Multivalent RBD formats can enhance B-cell activation due to the cross-binding of their receptors; however, the key role is played not by antigen multimerization but rather by the antigen design enabling better-exposed RBD epitopes [12]. For covalent dimerization, sequential tandems of RBD, and RBDs interconnected by N- or C-ends were used. Such dimeric forms elicited a 10- to 100-fold higher induction of antibody titers than the monomeric ones [12,13]. Meanwhile, vaccines based on non-covalent RBD dimers or by obtaining a fused RBD-Fc protein of human antibodies, and others were also proposed [36]. RBD trimers were obtained by introducing the fibritin trimerization domain of bacteriophage T4 into the C-end of RBD [14]. Not only did the dimeric and trimeric RBD variants induce high titers of neutralizing antibodies, but they also elicited stable immune responses of CD4+ and CD8+ T cells [37].

Multivalent RBD-based vaccines were obtained either by self-assembling particles whose monomers were fused with RBD using genetic engineering methods or by protein ligation technology when RBD was bound to the already assembled nanoparticles [15,16,38].

In our study, we combined the advantages of adjuvants and multivalent antigens and synthesized virus-like particles that exhibit RBD on their surface and contain yeast dsRNA inside. In the experiments on immunization of BALB/c mice, we found no statistically significant difference between the humoral immune responses in the groups of mice immunized with RBD+PGS, RBD + Al(OH)_3_, and RBD-PGS + dsRNA, while the group of mice treated with RBD without adjuvants had a two-order lower GMT of neutralizing antibodies and specific IgG compared to those indicated. Interestingly, in the group of mice treated with RBD-PGS conjugate without dsRNA, the specific humoral immune response to RBD was somewhat lower, thus emphasizing that adjuvants play a more significant role in our investigation than does antigen multimerization.

Since the induction of a specific T-cell response was previously shown for preparations/immunogens containing yeast dsRNA, we estimated the induction of specific T cells by RBD-PGS + dsRNA particles obtained in our study. ICS analysis of antigen-specific secretion of IFN-γ by CD4+ and CD8+ T cells and ELISpot analysis of specific secretion of IFN-γ in splenocytes of the immunized mice showed that the RBD-PGS conjugate combined with dsRNA elicited a strong specific T-cell immune response in BALB/c mice (Figure 3).

The effectiveness of using PGS in vaccines was previously demonstrated in the CombiHIVvac vaccine against HIV-1, which combines the conserved polyepitope immunogen approaches and the PGS microparticle concept. Immunogenicity and safety of the CombiHIVvac have been shown in preclinical studies in several animal species, and phase I clinical trial results obtained in human volunteers confirmed that the CombiHIVvac candidate vaccine was safe and did not cause side effects, and at the same time, inducing the HIV-specific humoral and cellular immune responses [39]. Therefore, we assume that the protective properties of the RBD-PGS + dsRNA particles obtained in this work will also be reproduced in humans, as it has been shown for CombiHIVvac and other antiviral vaccines based on nanoparticles.

The question of the contribution of post-vaccination humoral and cellular immune responses to protection against COVID-19 remains debatable. In the study of the mRNA-1273 COVE vaccine (Moderna) against SARS-CoV-2, neutralizing antibodies were not detected; however, the effectiveness of the vaccine was 50.8% [40]. Similarly, the Ad26.COV2.S vaccine (Janssen) showed 52% protection in clinical trials in South Africa, where there were many cases of infection with the beta variant, even though it did not generate high levels of antibodies against this variant of the virus [41,42]. At the same time, the Novavax vaccine showed a clinical efficacy of 86.3% against alpha (B.1.1.7) and 50% against beta (B.1.351) variants of SARS-CoV-2 in the absence of any noticeable CD8+ T-lymphocyte responses [43,44]. Sinovac and Bharat have also reported clinical efficacy in the absence of appreciable CD8+ T-lymphocyte responses [13,45,46].

The proposed vaccine combining the RBD protein and dsRNA from a killer yeast strain with polyglucin and spermidine not only combines the effectiveness of both humoral and cellular immune responses but also uses safe, biodegradable components. We suggest this can be an effective strategy for obtaining new protein-based vaccines against SARS-CoV-2.

In conclusion, it should be noted that the work has a number of limitations. For the analysis of the humoral response, it would be very useful to determine the antibody isotypes in the variety of specific antibodies that are induced after vaccination. This gives a more detailed picture of the humoral immune response. Also, to determine the specificity of antibodies, we used only recombinant proteins based on the Wuhan strain. It would be interesting to look at specific antibody titers using recombinant proteins based on the amino acid sequences of other variants such as Delta and Omicron. Similarly for virus neutralization, it would be better to use a wider panel of SARS-CoV-2 variants. To analyze the T-cell response, we limited ourselves to analyzing only the production of IFN-γ. A more complete picture could be obtained from the analysis of a wider range of cytokines and a more complete characterization of the phenotypes of specific CD4+ and CD8+ T cells.

## Figures and Tables

**Figure 1 vaccines-11-00808-f001:**
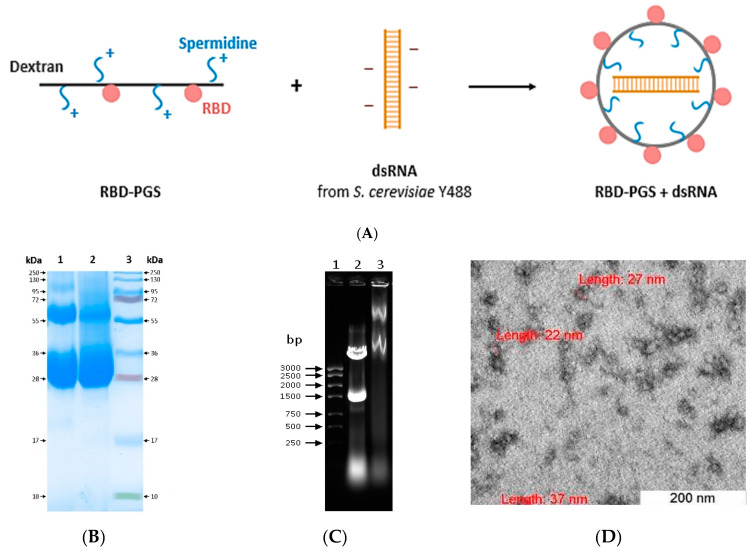
(**A**) The scheme for obtaining an RBD conjugate with polyglucin-spermidine complex and dsRNA. (**B**) Electrophoresis of RBD-PGS conjugate in 15% PAAG: (1) RBD protein, 40 µg; (2) RBD-PGS conjugate, 40 µg; and (3) molecular weight markers, 10–250 kDa. (**C**) Electrophoresis in 1% agarose gel: (1) molecular weight markers; (2) intact dsRNA; and (3) RBD-PGS + dsRNA particles. (**D**) An electron micrograph of RBD-PGS + dsRNA particles.

**Figure 2 vaccines-11-00808-f002:**
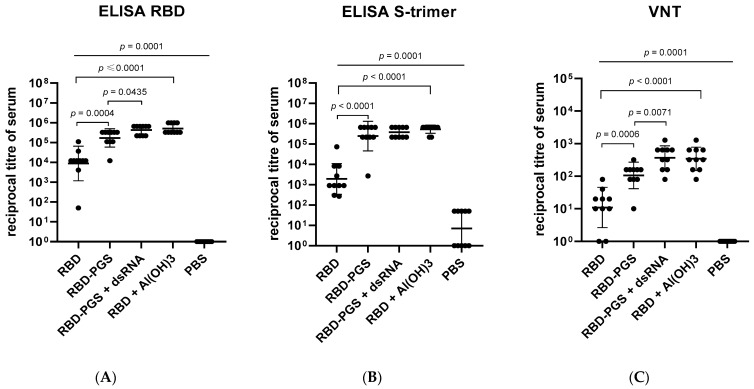
Immune response to the conjugated RBD in BALB/c mice: (**A**) Serum IgG specificity to recombinant RBD in ELISA; (**B**) Serum IgG specificity to recombinant S protein in ELISA; (**C**) Serum neutralization titers against SARS-CoV-2/Victoria/1/2020. The data were plotted as the geometric mean ± SD. Statistical analyses were performed using the Mann–Whitney U test and Kruskal–Wallis H tests. *p*-values indicate statistical significance.

**Figure 3 vaccines-11-00808-f003:**
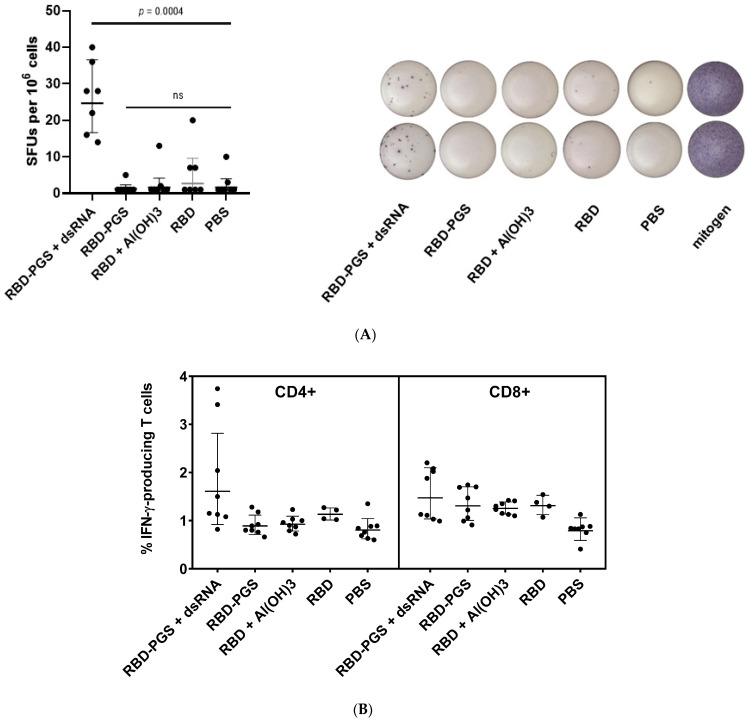
T-cell immune response to conjugated RBD in BALB/c mice: (**A**) ELISpot and (**B**) ICS analyses of mouse splenocytes. The geometric mean ± SD is presented. Statistical analyses were performed using the Mann–Whitney U test and Kruskal–Wallis H tests. *p*-values indicate statistical significance.

## Data Availability

The data presented in this study are available on request from the corresponding author.

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
