# Peer review of "SARS-CoV-2 RBD Conjugated to Polyglucin, Spermidine, and dsRNA Elicits a Strong Immune Response in Mice"

_vaccines, 2023, doi:10.3390/vaccines11040808_

Round 1

Reviewer 1 Report

SARS-CoV-2 RBD conjugated to polyglucin: spermidine and dsRNA elicits a strong immune response in mice. The article is well-balanced. Requesting authors to make the following changes;

Major comments

There are many punctuations errors so ensure those along with minor English check throughout the manuscript

Ensure that all the abbreviations are defined during the first instance of usage

In the introduction, authors should discuss about the variants of SARS-CoV-2 from   

https://doi.org/10.1002/jmv.27717 and https://doi.org/10.1038/s41579-021-00573-0

Minor comments

L- 72: “per se exerts”??

L-77-78: “The double stranded RNA forms are foreign to cells, and naturally represent either a viral genome or products of the viral reproductive cycle” - foreign to cells ??

L-79: TLR3 cellular receptor – Provide full form

L-85: type 1 IFN genes – Provide full form

Author Response

The team of authors would like to thank the reviewer for careful reading of the manuscript and important comments.

Major comments

There are many punctuations errors so ensure those along with minor English check throughout the manuscript

We carefully re-read the article again and tried to correct all punctuation errors.

Ensure that all the abbreviations are defined during the first instance of usage

We checked that all abbreviations encountered for the first time are defined.

In the introduction, authors should discuss about the variants of SARS-CoV-2 from   

https://doi.org/10.1002/jmv.27717 and https://doi.org/10.1038/s41579-021-00573-0

 Since the announcement of a global COVID-19 pandemic, there has been a continuous evolution of SARS-CoV-2, accelerating after about a year, leading primarily to a change in its antigenic properties [https://doi.org/10.1002/jmv.27717]. This circumstance reduces or completely cancels the effectiveness of vaccines developed at the first stage, using the sequences of the original virus (Wuhan strain) [https://doi.org/10.1038/s41579-021-00573-0]. However, along with the work on updating the antigenic composition, work is needed to improve vaccine platforms to increase their safety and increase efficiency.

Considering your remark, we have tried to more clearly express the main idea of the work - the study of a new vaccine platform. Also, we have described the limitations of our work, including the use of the Wuhan RBD variant.

Minor comments

L- 72: “per se exerts”??

This means that by itself, when used separately, it has an immunomodulatory effect.

L-77-78: “The double stranded RNA forms are foreign to cells, and naturally represent either a viral genome or products of the viral reproductive cycle” - foreign to cells ??

The double stranded RNAs are foreign to mammalian cells, and naturally represent either a viral genome or products of the viral replicative cycle.

L-79: TLR3 cellular receptor – Provide full form

Toll-like receptor 3

L-85: type 1 IFN genes – Provide full form

interferons

Reviewer 2 Report

The introduction is well-structured and provides a clear rationale for the study, highlighting the potential benefits of novel approaches to vaccine development. The methodology appears to be well-designed and comprehensive. In the results, the authors first described the successful preparation and characterization of RBD conjugate with polyglucin-spermidine and dsRNA constructs; the data are well presented. In Figures 2 and 3, they presented data on humoral and cellular immune responses to the conjugated RBD in BALB/c mice. Overall, the data are interesting. However, the authors should consider a better characterization of both humoral and cellular responses. Antibody isotype expression and a phenotypic description of CD4 and CD8 T cells involved using FACS staining would be helpful. The authors should also consider presenting data on other cytokines. The discussion is well-structured but should better cover the potential limitations of the study.

Author Response

The team of authors would like to thank the reviewer for careful reading of the manuscript and important comments.

The introduction is well-structured and provides a clear rationale for the study, highlighting the potential benefits of novel approaches to vaccine development. The methodology appears to be well-designed and comprehensive. In the results, the authors first described the successful preparation and characterization of RBD conjugate with polyglucin-spermidine and dsRNA constructs; the data are well presented. In Figures 2 and 3, they presented data on humoral and cellular immune responses to the conjugated RBD in BALB/c mice. Overall, the data are interesting.

However, the authors should consider a better characterization of both humoral and cellular responses. Antibody isotype expression and a phenotypic description of CD4 and CD8 T cells involved using FACS staining would be helpful.

To assess the humoral response, we used the ELISA method and virus neutralization. Unfortunately, we did not determine antibody isotypes. But we will definitely take into account this remark and in the continuation of this work we will carry out such analyzes. We also did not begin to determine a phenotypic description of CD4 and CD8 T cells involved using FACS staining. In future work, we will definitely pay attention to this.

The authors should also consider presenting data on other cytokines.

To assess the T-cell response, we decided to focus on interferon-gamma production. Since this is the main marker of the antiviral response. Unfortunately, we did not measure other cytokines. However, your remark is very important and we will definitely carry out such work.

The discussion is well-structured but should better cover the potential limitations of the study.

We have prepared a paragraph about limitations in our work.

Our work has a number of limitations. For the analysis of the humoral response, it would be very useful to determine the antibody isotypes in the variety of specific antibodies that are induced after vaccination. This gives a more detailed picture of the humoral immune response. Also, to determine the specificity of antibodies, we used only recombinant proteins based on Wuhan strain. It would be interesting to look at specific antibody titers using recombinant proteins based on the amino acid sequences of other variants such as Delta and Omicron. Similarly for virus neutralization, it would be better to use a wider panel of SARS-CoV-2 variants. To analyze the T-cell response, we limited ourselves to analyzing only the production of IFN-γ. A more complete picture could be obtained from the analysis of a wider range of cytokines and a more complete characterization of the phenotype of specific CD4+ and CD8+ T cells.

Round 2

Reviewer 1 Report

Now the work is fine.

Author Response

Thank you very much for your valuable comments.